# Decellularization of Dense Regular Connective Tissue—Cellular and Molecular Modification with Applications in Regenerative Medicine

**DOI:** 10.3390/cells12182293

**Published:** 2023-09-16

**Authors:** Krzysztof Data, Magdalena Kulus, Hanna Ziemak, Mikołaj Chwarzyński, Hanna Piotrowska-Kempisty, Dorota Bukowska, Paweł Antosik, Paul Mozdziak, Bartosz Kempisty

**Affiliations:** 1Division of Anatomy, Department of Human Morphology and Embryology, Wroclaw Medical University, 50-368 Wroclaw, Poland; 2Department of Veterinary Surgery, Institute of Veterinary Medicine, Nicolaus Copernicus University in Torun, 87-100 Torun, Poland; 3Department of Toxicology, Poznan University of Medical Sciences, 60-631 Poznan, Poland; 4Department of Basic and Preclinical Sciences, Institute of Veterinary Medicine, Nicolaus Copernicus University in Torun, 87-100 Torun, Poland; 5Department of Diagnostics and Clinical Sciences, Institute of Veterinary Medicine, Nicolaus Copernicus University in Torun, 87-100 Torun, Poland; 6Physiolgy Graduate Faculty, North Carolina State University, Raleigh, NC 27695, USA; 7Prestage Department of Poultry Sciences, North Carolina State University, Raleigh, NC 27695, USA; 8Department of Obstetrics and Gynecology, University Hospital and Masaryk University, 601 77 Brno, Czech Republic

**Keywords:** tendon, ligament, bioscaffold, extracellular matrix

## Abstract

Healing of dense regular connective tissue, due to a high fiber-to-cell ratio and low metabolic activity and regeneration potential, frequently requires surgical implantation or reconstruction with high risk of reinjury. An alternative to synthetic implants is using bioscaffolds obtained through decellularization, a process where the aim is to extract cells from the tissue while preserving the tissue-specific native molecular structure of the ECM. Proteins, lipids, nucleic acids and other various extracellular molecules are largely involved in differentiation, proliferation, vascularization and collagen fibers deposit, making them the crucial processes in tissue regeneration. Because of the multiple possible forms of cell extraction, there is no standardized protocol in dense regular connective tissue (DRCT). Many modifications of the structure, shape and composition of the bioscaffold have also been described to improve the therapeutic result following the implantation of decellularized connective tissue. The available data provide a valuable source of crucial information. However, the wide spectrum of decellularization makes it important to understand the key aspects of bioscaffolds relative to their potential use in tissue regeneration.

## 1. Introduction

Connective tissue (CT) is an omnipresent and versatile tissue representing a wide spectrum of functions. One of the functions is forming attachments between components of motor skills through creating ligaments, tendons, fasciae and aponeurosis. These four structures are representatives of dense regular connective tissue (DRCT), which is tissue rich in collagen fibers. The regeneration of DRCT is challenging because of the molecular composition of those structures. Its relatively low cell-to-fiber ratio favors disorganized fibrotic scarring around the wound [1]. It may increase the risk of reinjury committed by the poor mechanical qualities of this method of wound closure [2]. Poor vascularity and low metabolic rate also disturb efficient regeneration [3,4]. Current therapies employ plenty of grafts and sutures to reconnect the tissues, especially ligament/tendon-to-bone prostheses [5], grafts [6] or cavity sealing (hernia meshes) [7].

One of the potential therapies is the tissue-specific wound dressings obtained by tissue decellularization. Removing cells produces an immunologically neutral material with protein composition and fibers organization as in the target tissue that can be used as an efficient and safe wound dressing, also in xenogeneic transplants [8]. The decellularized tissue is composed of fibers and a ground substance that make up the extracellular matrix (ECM), which is the crucial factor in determining the mechanical properties of tissue. It also fulfills the function of a scaffold for surrounding cells, especially fibroblasts, which are responsible for ECM synthesis. Therefore, the safety of the ECM during decellularization is an essential part of protocol optimization. Many decellularization methods and techniques have been described, but there remains an absence of studies comparing various protocols. The regenerative potential of obtained material can even be increased by reseeding the ECM with cells [9] or supplementing with proteins or growth factors [10]. Due to the wide possibilities of composition and shape modification, this solution represents a promising technique for DRCT regeneration. Many aspects of tissue decellularization and their usage are still unknown. The current state of knowledge constitutes a valuable resource of information that needs to be organized. It is important to consider DRCT decellularization knowledge in terms of the efficiency and safety of the techniques.

## 2. Essential Morphological and Molecular Features of Dense Regular Connective Tissue (DRCT)

The physical properties of the ECM are mainly determined by the number and distribution of fibers. The molecular composition of proteins in the ECM is dominated by collagen fibers, especially type I collagen fibers. In DRTC, it establishes over 80% of the tissue dry mass [11]. Furthermore, the main part of the remaining proteins are elastin, proteoglycans and glycosaminoglycans [12]. The expression of some kinds of proteins can vary from region to region, even within a single structure [13]. ECM remodeling relies on collagen degradation by matrix metalloproteinases (MMP), universally occurring in many tissues [14]. Fibers are mainly organized parallel to a main functional axis, especially in tendons and ligaments (Figure 1). The fasciae fibers are organized in sublayers where fibers are oriented in a longitudinal pattern [15]. The direction of the fibers depends on the direction of the tensile forces acting on the structure [16]. This distribution results in immense mechanical strength, maintains tissue integrity and provides tensile stiffness. Disturbances in the orientation of collagen fibers can weaken the structure, exposing the tissue to injuries [17]. There are noticeable differences in the thickness and density of the fibers depending on the localization of the structure, which is a consequence of the unique cross-linking profile of each tissue [18]. There are some other diversities between the individual fibrous structures in the molecular composition of collagen, e.g., in post-translational hydroxylation of proline and lysine [11]. These amino acids are enzymatically modified into hydroxyproline and hydroxylysine, which is crucial in collagen triple helix folding [19]. In this process, plenty of enzymes are involved, such as those in the lysyl hydroxylase (LH) family and prolyl 3-hydroxylase (P3H) family [20]. Reduced hydroxylation activity results in impaired collagen stability, fibers organization and diameter [21]. Moreover, products of hydroxylation promote differentiation and migration of tendon cells in vitro [22].

Fibroblasts synthesize a wide spectrum of proteins, and they are the main architects of the synthesis and remodeling of the ECM structure [13]. One of the specific types of fibroblasts is the tenocyte type, with the tenocyte being a main cellular unit of tendon. Morphologically, tenocytes are longer and more slender than fibroblasts, but functionally they are very similar [23]. Tenoblasts, which are immature tendon cells, are relatively round cells with large ovoid nuclei [24]. They are motile and highly proliferative and are dominant in young tendons [25]. Tenoblast sources are tendon stem cells (TSC), which differ morphologically from mature cells in shape, assuming a cobblestone-like arrangement with smaller cell bodies and larger nuclei [26]. Stecco et al. [27] suggest that, in the fascia composition, are specific cells—fasciacytes. Using hematoxylin and eosin staining, they are visible as enlarged and clearly round cells, different from the fascial fibroblasts [27]. They are located along the surface of each fascial sublayer and specialize in hyaluronan expression—a factor facilitating fascial–muscle gliding. Regular elongated fascial fibroblasts are not immunoreactive for fibroblast-specific protein 1 (FSP1; S100A4) [27], which explains the coexisting different types of fascial fibroblasts noted by Thankam et al. [28]. All of the noted cell types demonstrate expression of specific fibroblast markers, such as vimentin (*VIM*) or platelet-derived growth factor receptor alpha (*PDGFRA*), as shown in Table 1. There are many other cell colonies with different gene expression profiles in connective tissue composition [29].

**Figure 1 cells-12-02293-f001:**
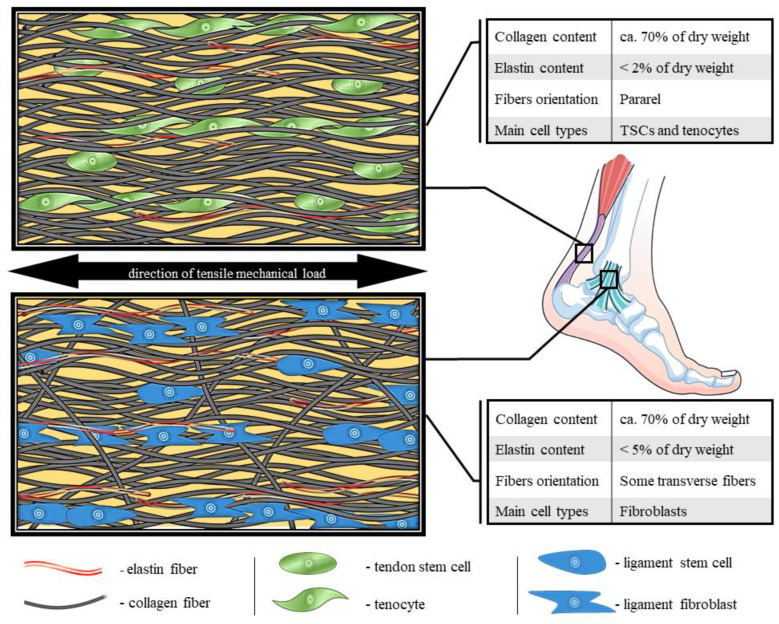
Comparison of molecular structure, fibers organization and composition of tendon and ligament [24,30,31,32,33,34,35].

Due to the ubiquity of DRCT tissue, it suffers in many injuries to and pathologies of the musculoskeletal system. The most common injuries of DRCT are Achilles tendon rupture [47], injury to the ligaments of the knee joint, meniscus damage [48] and hernia [49]. Full recovery of injured DRCT includes three stages of healing: the inflammation, proliferation and remodeling stage. During the first 48 h, necrotic debris of injured DRCT is digested, and pro-inflammatory molecules are expressed. In the second phase, the dominant process is the proliferation of fibroblasts and tenocytes and the synthesis of collagen III, IV and XII, which are less durable than collagen I, which is dominant in DRCT. The stage is characterized by the highest expression of DRCT growth factors [50] and remodeling molecules, especially of the MMP family, which is also maintained in the third phase of regeneration [51]. Lastly, the remodeling phase is initiated 3 weeks after injury. Elongated cells and collagen fibers undergo maturation and reorientation along the direction of mechanical stress. The newly synthesized fibers transform into collagen I. The tissue remodeling process is the longest stage of DRCT regeneration. It is commonly mentioned that it can last up to 12 months after injury [52]. The healed area has a scar-like appearance, and the tissue is biomechanically less durable than the native tissue.

A crucial element in the tendon/ligament/fascia functioning is enthesis at the point of its junction to the bone, shown in Figure 2. One of the types of connection is fibrocartilage enthesis, mainly located on epiphyses and apophyses of bones (i.e., Achilles tendon to the calcaneal tuberosity). This junction is composed of four layers: DRCT with a parallel arrangement of collagen fibers, a fibrocartilaginous area with chondrocytes, a region of calcified fibrocartilage and bone [53]. The second type is fibrous enthesis, where DRCT is fused with bone through ossificated collagen fibers. It is mostly located at the metaphysis and diaphysis of long bones (i.e., adductor magnus to the linea aspera) [54]. The main part of development of these areas is processed during embryonic and postnatal growth. In ossification of junctions, regulation of *SOX9* and *GLI1* genes is involved. Because of bone elongation, some junctions have to migrate to reach the target position on mature bone. During entheses, the migrating population of Sox9+ cells are replaced by Gli1+ cells which form mature connections. In stationary junctions, Sox9+ cells inhibit the enthesis structure during the whole process of bone elongation and ossification, with only some Sox9+ cells expressing *Gli1* [55]. Increased ossification is associated with inhibition of tendon/ligament development [56], which is disturbing for efficient DRCT regeneration. It may also be a reason for the formation of heterotopic ossification, which disrupts the DRCT properties [57]. Stiffening and weakening of the collagen fibers can also be caused by intense tensile forces [58], which may lead to hernias. The durability of the fascia plays a key role in balancing the level of MMP and their inhibitors [45] and also in cooperation of circulating MMP and procollagen [59]. One of the challenges is also the interruption of abdominal wall fasciae/aponeuroses continuity as a result of the operation, which also faces regeneration problems [60]. 

In conclusion, DRCT regeneration requires comprehensive support. It is especially necessary to concentrate on the aspects of providing the undisturbed orientation and distribution of collagen fibers, efficient inter-tissue connections and preventing ossification and fibrous scarring.

## 3. Decellularization Methods

The most important aspects of the decellularization process are its efficiency of cell extraction and preservation of the ECM fibrous structure. Universally described criteria for the safety and efficiency of decellularization are DNA and nucleus residues. Crapo et al. suggest implementing three quantitative indicators of effective decellularization: (1) less than 50 ng dsDNA in 1 mg of ECM dry weight; (2) residual fragments of DNA shorter than 200 bp; (3) lack of visible nuclear material in tissue sections stained with 4′,6-diamidino-2-phenylindole (DAPI) or hematoxylin and eosin [61]. Appropriate cell extraction will protect the transplanted graft against an immune response, which is the base of graft survival, and, finally, effective tissue regeneration [62]. In relation to its function and localization, DRCT is exposed to strong tensile forces, which it must resist. The decellularization retains an unchanged content of Hyp in the ECM, which is an indicator of the mechanical resistance of collagen [63,64]. There are no reports of the effect of decellularized tissue on hydroxyproline and hydroxylysine synthesis during the tissue regeneration process. Injury of a tendon, ligament or fascia, especially in athletes, puts returning to preinjury functionality at risk [65,66,67]. The limbs with healed ligaments or tendons achieve lower results and asymmetricity in athletic performance [68,69]. Tendons/ligaments resistance requires directional stretching forces for proper development. In the embryonic state, it takes place in the safe environment of the uterus [70]. At the adult stage, the healed area is commonly immobilized, which does not provide the necessary tensile forces to properly reorganize the ECM. Furthermore, when the limb is not immobilized, the regenerating area is exposed to reinjuries as a result of too intense tensile forces. The effect of ECM disorganization during the tissue regeneration may be minimalized by preservation of native fibers’ structure and composition. 

The DRCT decellularization protocols exhibit plenty of different techniques, shown in Table 2, which require using a great variety of components and actions. However, most often, protocols use the cooperation of multiple types of agents, which makes it difficult to compare effectiveness of individual components of a protocol and their safety. Basic division consists of physical and chemical methods.

### 3.1. Physical Methods

#### 3.1.1. Electroporation

Electroporation-induced decellularization employs high-magnitude electric pulses to reduce integrity of the cell membrane. The electric field crossing through the cells destabilizes the electric potential of cell membranes and results in the formation of nanoscale pores. Homeostasis of cells is destabilizing, which is the direct cause of cell death. Modifications of the voltage or pulses frequency allow adjustment of the decellularization conditions in relation to the desired effects. The most commonly used type of electroporation is NTIRE (non-thermal irreversible electroporation), which minimizes thermal damages of the ECM [92]. This technique is rarely used. Some researchers apply it in various tissues because of its safety and simplicity, which make it possible to process in situ [93]. However, this technique is characterized as insufficient and causes increased inflammation in in vivo analysis [94]. The technique is used mainly to implement autologous grafts or in situ procedures, which are performed even on DRCT, while maintaining full strength efficiency of the structure [95,96].

#### 3.1.2. Sonication

Sonication-induced decellularization uses high-frequency sound waves to form and collapse microbubbles to generate shear forces, microstreaming and shock waves. These forces disrupt cell membranes, leading to the release of cellular contents, and also induce ECM net looseness. The induction of waves can be generated using a direct sonicator or an undirected ultrasonic bath. The use of the indirect method, due to its global effect, enables the decellularization of larger samples and exhibits improved perseverance of the ECM structure but requires more time [72]. In the context of decellularization, sonication is an inefficient technique, but it is used with some other agents (i.e., chemical agents) [97] in promoting the release of cellular contents and facilitating the removal of cellular debris. Sonication-based decellularization protocols are applied to various tissue types, including blood vessels [98] and nerves [99], but there is little information about using it in DRCT. Despite loosening the structure of the ECM network, sonication does not contribute to reducing the strength of the decellularized tissue or collagen content [98]. On the contrary, increased gaps between the fibers may promote efficient recellularization and finally regenerate the tissue [100].

#### 3.1.3. Freeze–Thaw

The procedure of freezing samples induces intracellular ice crystal formation, causing mechanical disruption of cell membranes, leading to cell death. This technique most often involves the manual use of liquid nitrogen and an incubator, but there is a possibility of automating the process, which can be used in manufacturing processes [101]. The freeze–thaw method has been successfully applied to various tissues and also DRCT, including large tendons and ligaments [63,73]. Unlike other physical techniques, FT has been found to be an effective, independent method of decellularization. Post FT, much cellular debris remains inside the ECM, but it appears to be a more efficient tool for cell extraction than popular chemical agents [74]. However, FT applied with chemical methods can result in a protocol with an efficiency of up to 95%, which makes the decellularized tissue immunologically neutral and safe in xenogeneic transplants [74]. Appropriate rinsing of the tissue decellularized by FT makes it completely cytocompatible and provides a solid scaffold for recellularization [102]. Decellularized tissue also preserves the strength and mechanical properties of the native tissue [63,74].

#### 3.1.4. High Hydrostatic Pressure

High Hydrostatic Pressure (HHP) involves applying elevated pressure to a sample immersed in a liquid medium. High pressure leads to irreversible loosening of the phospholipid bilayer, which causes cell membrane permeability and ultimately results in cell death. Using an HHP-involving methodology, the periodontal ligament was successfully decellularized, resulting in a cell-free ECM with high mechanical strength and low residual DNA content, suggesting the technique can be used as an efficient decellularization tool [91]. The decellularization technique is better described for other tissues, i.e., uterine or aortic samples, where HHP shows an efficiency in removing cells as high as that of popularly used chemical agents [75,76] without persistent cytotoxic effect. Crucial advantages of HHP are minimal protein denaturation and preservation of the mechanical resistance of native tissue. Due to the mechanism of action, HPP proceeds independently of the sample’s size and structure so it does not require advance optimization. Most commonly used conditions for HHP require 980 MPa pressure, but efficient decellularization proceeds at a value of 200 MPa [103]. Cell extraction can be improved by using cyclic HHP with repeatable high and low pressures on the sample [104]. Despite the high universality of the method, it requires comprehensive optimization and validation in DRCT. The high cost associated with this technology is mentioned as a disadvantage.

In conclusion, a significant portion of physical methods, especially sonication and electroporation, demonstrate low efficiency of cell extraction, and they should constitute only a stimulus initiating a decellularization process in a multi-step protocol. The procedure with them should be enriched with other cell extraction agents. Sonication, although not well studied in the context of DRCT, leads to significant loosening of ECM fibers and may be precluded from use with this tissue. Electroporation, despite promising results, has not been sufficiently evaluated for its effects on DRCT. All of the above methods do not require extensive specialized equipment, and FT shows the most promising results.

### 3.2. Chemical Methods

#### 3.2.1. Triton-X 100

A commonly utilized non-ionic surfactant used as a component of decellularization protocols is Triton-X 100. It extracts the cells through lipid bilayer disruption, a crucial factor in the breakdown of cell membranes and subsequent release of cellular contents. Triton X-100 should preserve protein structure and maintain the structural integrity and composition of ECM. A significant relaxation of collagen fibers occurs in Triton-X-treated tendons [77]. Researchers have also indicated the presence of a few residual tendon cells, suggesting insufficient decellularization, which is confirmed by other studies [78,79]. Much more efficient decellularization may occur using a combination of Triton X-100 and other agents, such as FT [105], SDS [106] or trypsin [107], which have been proved in other tissues. Most protocols indicate the use of a 1% concentration of Triton X-100 solution [108]. However, as evidenced by Luo et al. [109], a concentration as low as 0.1% will ensure effective cell extraction. Additionally, the time of incubation is crucial, especially for ECM organization and composition, because long-term reactions may cause protein disruptions [109]. Despite this, the use of Triton X-100 does not significantly weaken the strength and physical properties of the tissue [77].

#### 3.2.2. Sodium Dodecyl Sulfate (SDS)

The ionic detergent SDS is one of the common supplements in the decellularization protocol. Its mechanism of action is based on the dissolution of cell membranes and also on disruption of protein interactions and structure. SDS is an extremely effective molecule for decellularization but is also dangerous for the surrounding proteins [80]. Applying even 0.1% SDS will disturb fibers organization, and 1% will produce an amorphous structure lacking a fibrous net [81]. SDS treatment leads to higher denaturation of collagen than in the case of Triton X-100 [110], which affects tissue [74]. The use of SDS has a long-lasting cytotoxic effect [111]; therefore, decellularization requires consideration of thorough and advanced washing [112,113]. SDS is often used at low concentrations as a supplement to the DRCT decellularization protocol [102,114].

#### 3.2.3. Sodium Azide

Sodium azide is a well-known inhibitor of cytochrome C oxidase in the mitochondria. It causes disruption of cellular function and cell death by reducing ATP synthesis. It is a commonly used component of DRCT decellularization protocols, acting as a bacteriostatic agent [82]. Due to its high cytotoxic effect, its efficiency depends on the activity of cells; therefore, it has the greatest cell extraction effect on cancer cells [83]. Decellularization protocols for DRCT based on sodium azide have not been reported; however, due to its low cell activity, it may have low efficacy.

#### 3.2.4. Latrunculin B

The functionality of Latrunculin macrolide produced by the marine sponge relies on depolymerization of actin filaments, which are one of the major structural components of the cell cytoskeleton. It is also a major ingredient of the structure of muscles, which is a reason why that the agent is commonly used for muscle decellularization. Furthermore, Latrunculin B was described once as a valuable agent of safe tendon cell extraction [84]. The advantage of this method is that the action is directed at the cells, leaving the native ECM network structure and an unchanged protein composition, which has been proved in muscle tissue [115]. In comparison with commonly used agents, Latrunculin B shows higher efficiency than Triton X-100 and greater safety than SDS [85], but its action is still not well known in DRCT; however, it appears to be a promising technique.

#### 3.2.5. EDTA

Ethylenediaminetetraacetic acid (EDTA) is a chelating agent whose mechanism of decellularization is based on separating the cells from the surrounding ECM by binding to divalent metal cations at the site of cell adhesion. Protocols using EDTA alone are described as insufficient [86]. Therefore, EDTA is commonly applied in combination with trypsin [97,116]. As a supplement to the decellularization protocols in various tissues, EDTA increases the cell extraction efficiency of the physical method but may reduce the amount of protein in the sample [87]. Otherwise, EDTA does not have a significant impact in the decellularization process based on sodium deoxycholate and deoxyribonuclease (DNase) [117]. The main utility of EDTA in decellularization is intensification of trypsin action which, alone, leads to the formation of cell aggregates [118].

#### 3.2.6. Trypsin

Trypsin is a pancreatic enzyme that shows robust enzymatic activity in digesting membrane proteins, often leading to cell death. The mechanism of trypsin action is based on targeting the bonding with the C-terminal side of arginine and lysine, resulting in cleaving peptides. Trypsin treatment should be precisely balanced between effective decellularization and protein preservation in DRCT. Incubation of porcine myocardial tissue for 5 h [119] or porcine carotid arteries for even 1 h [88] with 0.5% (*w*/*v*) trypsin caused significant slackness of ECM fibers. Therefore, the enzyme is commonly used in multi-step protocols with other agents, especially in DRCT, where ECM preservation is a priority [8,120]. In bovine flexor tendons, the use of only a 0.05% trypsin concentration, in combination with DNase I, provided efficient decellularization while preserving the native structure of the fibrous net [89]. The purity of the trypsin solution plays a key role. According to the European Medicine Agency [121], the content of various types of proteases is variable between lots of trypsin product series. The presence of other proteases reduces the specificity of the solution, increasing the efficiency of cell extraction. One of the impurities may be the presence of chymotrypsin, which presents the abilities to cleave proteins and extract cells [122].

#### 3.2.7. Nucleases

DNase and RNase are endonucleases whose role is to cleave DNA or RNA in the extracellular space. Important for DNase activity is the presence of calcium and magnesium ions [123]. Due to the nuclease’s mechanism of action, its role in decellularization is to complete and expedite the process of removing DNAs and RNAs from the cells extracted by other agents. Nucleases prevent nucleic acid debris from sticking to the ECM, accelerating the preparation process and improving cytocompatibility [96]. Using DNase on porcine esophageal tissue led to the removal of over 95% of DNA remaining after processing with other chemical agents [90]. In addition, during DRCT decellularization, addition of DNase increases the protocol efficiency with sufficient preservation of the ECM structure and collagen content [124]. The preservation of the ECM structure after DNase digestion was also demonstrated in periodontal ligament decellularization [73].

A plentiful number of scientific papers describe the action of decellularization agents. However, there is a dearth of direct comparisons of procedures in terms of DRCT treatment. Various tissues respond differently to the same protocol of decellularization [64], which makes it crucial to analyze and compare the effects of all procedures on DRCT. Some of the above show promising results and should be precisely tested in the context of composing a multi-step decellularization protocol. The engagement of different mechanisms of action in a multi-agent protocol could maximize cell extraction efficiency using lower concentrations of chemical factors. Reducing the use of detergents will minimize the negative effects of ECM, especially disturbances in the fibers net and cytotoxicity.

## 4. Modifications of dECM Scaffolds

### 4.1. Cell Seeding

Decellularized ECM (dECM) can be used in various ways, as shown in Figure 3. Depending on the requirements and possibilities, dECM can be used as an acellular or in vitro recellularized graft. Cell colonization of acellular graft will proceed in vivo after implantation into the body with cells from surrounding tissue. dECM of rabbit abdominal fascia (flat, 3 × 3 cm), after implantation, was completely reseeded after approx. 2 weeks, and there appeared to be no significant differences between graft and native tissues [8]. In case of the larger structures, i.e., the rabbit gastrocnemius muscle tendon, after 2 weeks, the acellular graft was reseeded only peripherally. After 8 weeks, the graft was still noticeable but had significant progressive integration with native tissues [125]. Decellularized tendons are also a great solution for xenografts in ligament regeneration. A comparison between an acellular xenograft and cellular allograft (with no decellularization) showed that regeneration of the anterior cruciate ligament (ACL) proceeded similarly in both strategies, with complete tissue integration after 26 weeks. Additionally, mechanical properties of regenerated ligaments were comparable [126]. Using a cell-free implant requires less time for preparation, and the procedure is less complicated than using a recellularized graft. However, a dECM coated with cells exhibits more efficient regeneration and requires less time to integrate with native tissue [127,128]. Among the latest research, the most commonly used cells in DRCT recellularization are adipose-derived stem cells [129] and tendon-derived stem cells [116]. Recellularization can be preceded passively by incubation of the dECM in a vessel with a cell culture or actively by placing the cells inside the dECM using various techniques. The passive method is the most commonly used, but it shows low cell penetration into the dECM [74]. Active placement of the cell solution inside the dECM using a needle seems to be more efficient because of the higher number of cells infiltrating between ECM fibers soon after injection [130]. However, a direct comparison proved that the early high content of cells is only temporary, and the cell solution will leak quickly. After 5 days of incubation, both techniques showed a comparable viability and count of cells [131]. Reseeding can also be performed with flowing systems [128], but there is little information about using this method in DRCT. The efficiency of recellularization can also be increased by applying the physical methods of decellularization [74]. It is related to loosening the fibrous net and making it easier to infiltrate the ECM with cells. The efficiency of cells reseeding is also strictly dependent on the shape because the flat form of dECM simplifies cell access to the entire volume of the collagen network.

### 4.2. Native Structure

The structure of dECM can be modified in many possible ways. One of the possibilities is to leave the native structure and shape of the collected sample. This solution is a common way of processing ligament and tendons from small animals [132], where the only integration into the shape of the sample is cuts during sample harvesting. In the case of large tendon or ligaments, a commonly used modification is cutting a sample to establish shape and size for the decellularization process [74]. This shape and size adaptation is also crucial for grafting other DRCT structures, such as the fascia [8]. One of the most advanced methods of shape remodeling with preserved native ECM cross-linking is the method used by Song et al. to regenerate rat Achilles tendon. Flat, sectioned slices of dECM were reseeded with TSC and then rolled into a rounded shape. Due to the flat shape of the pieces, recellularization proceeded quickly and efficiently, and the rolling made it possible to mimic the native shape of the tendon [9]. This reseeding procedure also solved the problem of long, peripheral cell colonization of the massive structure of tendons and ligaments.

### 4.3. Structure Modifications

Other strategies rely on the disruption of the primary cross-linking during the remodeling of the dECM form. A common transformation of the dECM is the formation of a hydrogel. The first stage of its production is lyophilization and fragmentation of dECM into a powder [133]. The next step is homogenization by digestion in pepsin [134] or acetic acid [135] to form a pre-gel. Final gelation is processed by incubation at 37 °C. The processed scaffold shows high cell reseeding properties as well as unlimited possibilities for forming the shape of the implant without changing the protein composition of the ECM [136]. This solution is also promising in the case of regeneration of tissue connection areas. Treating the ruptured enthesis with a tendon–bone interface scaffold increases the efficiency of the healing process with rapid restoration of the mechanical properties of the junction [134]. One of the aspects of proper healing is the tendon dECM’s ability to promote expression of paxillin, a marker of muscle–tendon junctions [116]. A hydrogel can be also formed as a flat wound dressing by drying homogenized dECM on a flat plate. Covering a ruptured tendon with acellular flat dressing not only increases the efficiency of regeneration, but also prevents tendon adhesion to nearby tissues in the healing process [135]. An innovative method of forming dECM is thread spinning, which offers potential opportunities for sewing ruptured DRCTs using sutures with the composition of native tissue. Lyophilized bovine tendons were separated and then rolled into a linear material whose mechanical strength was similar to silk [89].

### 4.4. 3D Bioprinting

Forming the scaffold shape can also be achieved by 3D printing technology. This technique uses a pre-gel with or without cells which is cross-linked during the time of printing. The most commonly used technique of printing with dECM hydrogels uses a head with a mechanical extrusion and incubating system for pre-gel staining. Toprakhisar et al. used a system with a glass capillary tube, aspirating the pre-gel from a cool reservoir and incubating it at 37 °C for 6 min to transform the liquid into a gel. After incubation, the gel was extruded from the glass capillary on the printing area [137]. This process can be improved by appropriate optimization of the dECM bio-ink preparation. A shorter time of dECM digestion results in a more viscous bio-ink [138]. This improves the possibility of forming bio-ink and also shortens the time of cross-linking, which allows the use of only a heated work table for the incubation process. The 3D bioprinting method works well in in vivo analyses. The implant composed from decellularized tendon and bone with cells promoted effective regeneration of murine rotator cuff tendons, preserving full functionality and mechanical properties [139]. Printing properties can also be improved by supplementing the pre-gel with various reinforcing agents such as gelatin [140].

### 4.5. Enrichment of the Scaffold Composition

Considering the most important aspects of DRCT treatment, the efficiency of wound dressings and implants can be increased by manipulating composition with supplements. Biomaterial can be supplemented at any point of time during the preparations. Ingredients can be added during the decellularization protocol [141], after decellularization through incubation in a supplement solution or perfusion [142] or during pre-gel preparation [143]. Scaffolds can be enriched regardless of the structure. One of the important aspects of healing DRCT elements is mechanical resistance. Some strategies were applied to various tissues. Addition of polyethylene glycol diacrylate (PEGDA) to an annulus fibrosus decellularization protocol improved the compressive strength of biomaterial [143]. Cytocompatible polymer PEGDA has the ability to cross-link with the dECM collagen network and strengthen the dECM structure. The possibility of increasing the cross-linking potential of collagen fibers is also shown by oxidized naringin [144], which also increases basic fibroblast growth factor (*BFGF*) expression during DRCT healing. Supplementation of l-ascorbic acid (vitamin C) has a significant impact on collagen production and also on mechanical resistance of DRCT [145]. Coating dECM with chemical elements (e.g., titanium, nitrogen, potassium) improves resistance to cracks generated under mechanical loads in bone [146]. Evidence that bone-derived dECM is an efficient matrix for ligament/tendon regeneration [147] suggests that similar use of inorganic elements may have a positive effect on DRCT regeneration.

Oxidative stress (OS) is a forceful disturber of proper DRCT recovery. Excessive production of reactive oxygen species causes the reduction of fiber production and the weakening of tendons and ligaments [148]. The presence of OS induces cell death, decreases proliferation and metabolic activity of cells and also increases *MMP* expression [149]. There are some molecules that have been suggested as efficient antioxidative agents which also promote cell proliferation and viability, such as curcumin [150], nicotinamide mononucleotide [151], β-lactoglobulin [152] and N-acetylcysteine [149].

The crucial aspect of any regenerating tissue is cell differentiation, which also can be improved by dECM supplements. In equine tendon samples, differentiation was assured by Transforming Growth Factor Beta 3 (TGFβ3), which led to tenogenic differentiation of adipose-derived stromal cells (ASCs) [153]. Growth factors can be also used for entheses healing purposes [154]. Especially in the areas of fibrocartilaginous entheses where proper reconnection of three types of tissue is essential for healing. Treatment with dECM increases the level of *SOX9* expression in differentiating cells of injured tissue, affecting the development of cartilage tissue, potentially contributing to the regeneration of entheses [155]. Expression of *SOX9* during cartilage healing is even higher when using tendon dECM [156]. However, this effect has not been studied in the context of DRCT healing. 

Extracellular vesicles (EVs) seem to have a comprehensive supplementary effect. The cargo of EVs includes lipids, nucleic acids and proteins, with a broad spectrum of activity [28]. Exosomes, a type of EV derived from TSC, promote proliferation, migration and expression of tendon-specific markers [157]. This effect on tenocytes can also be provided by exosomes of other cells, such as bone marrow mesenchymal stem cells (BMSCs) [158] or ASCs [159]. Supplementing the scaffolds with exosomes also promotes the healing of injured tendons by supporting fiber arrangement and anti-inflammatory activity and supporting biomechanical resistance in vivo [157,159]. EVs are crucial also when they are isolated from other sources. Platelet-derived EVs seeded on synthetic scaffolds also influence the regeneration potential of the tenocytes. It increases the expression of tendon-specific markers, improves ECM remodeling and also promotes expression of anti-inflammatory cytokines [160]. Seeding platelet-derived EVs on a collagen-coated synthetic scaffold improves the tenogenic potential of ASCs and deposition of tendon-like ECM and prevents shrinkage of the collagen fiber network in vitro [161]. It appears that the role of the platelet-derived EVs is to universally control cell differentiation such as osteoblast [162] or lymphocyte differentiation [163] across different tissues. 

After the decellularization process, despite cell extraction, most of the extracellular molecules remain in the dECM. These molecules, among others, take part in the aspects discussed above, such as the proliferation, differentiation or inflammation process. Nevertheless, supplementation can be an effective way to increase or decrease the intensity of some processes and to boost efficiency of injuries healing. Additionally, it can ensure proper regeneration when healing proceeds with a scaffold obtained from a source other than the target tissue.

## 5. Therapeutic Potential—Current Preclinical Success

The current state of the research advancement of the DRCT decellularization process enables the use of bioscaffolds in preclinical studies and, above all, in in vivo trials. Due to the functions and localizations of DRCT structures, the experiments are divided into two parts. One of the parts is the research about tendons and ligament, which have a similarity in shape and composition that allows application of the process interchangeably in tendon/ligament healing [164]. Another part of the research is experiments on flat DRCT structures such as fasciae and aponeuroses. 

There is a significant disproportion between these two areas of research, as there are few studies on decellularized flat DRCT. Abdominal wall reconstruction is commonly performed with synthetic meshes, replacing native tissues [165]. As proved by Buell et al., grafts obtained from fascia dECM demonstrate equivalent mechanical resistance to commercially available mesh [166]. Subcutaneously grafted muscle fascia dECM demonstrates progression of collagen deposition, neovascularization and efficient cell colonization without evidence of inflammation or rejection. Closure of a rabbit abdominal wall full-thickness defect with a fascia xenograft also presented positive recovery. After transplantation, the animal did not demonstrate any signs of infection or graft rejection, with the graft achieving recellularization and integration with native tissue. The graft had a long-term effect because, 6 months after the operation, the animal still did not suffer hernias, bleeding or other local complications [8].

Biomechanical aspects of tendon/ligament provide promising results; the bioscaffolds show similar strength and stiffness to the native tissue [167,168]. In comparison with synthetic, commercially obtained scaffold, tendon-derived porcine dECM exhibited more effective cell colonization and less inflammation in vivo in a mice model. Analysis after 12 weeks showed that DNA content on implanted cell-seeded scaffold was compatible with the samples before decellularization, which is proof of the effective regeneration potential of dECM [82]. Regeneration potential was also proven in ruptured rabbit Achilles tendon, implanted with autograft and xenogenic dECM seeded with BMSCs. Both strategies exhibited efficient tissue healing with collagen fibers synthesis and biomechanical parameters similar to the native tissue. All animals after 10 days returned to full motor skills without any complications, which indicates tendon dECM as an effective therapeutic strategy in animal models [169]. The lack of differences between the efficiency of xenogenic and allogeneic implants was also proven by the complete regeneration of the ovine ACL [126]. Complete regeneration was also observed in the case of cell-seeded dECM implantation in the injured rabbit’s rotator cuff. During recovery, animals did not lose body mass and kept a full range of limb movements [167].

## 6. Conclusions

The high collagenic composition of DRCT structures ensures an impressive mechanical resistance but also contributes to low metabolic activity and poor regeneration abilities. A promising method to replace ligaments seems to be using implants, which also faces difficulties. The best results appear to be with autogenic implants [47], but they require a complicated and long-lasting sourcing process. Commercially available synthetic implants often do not have a pro-regenerative effect and only have a passive role in connecting injured tissues.

Decellularization of DRCT structures seems to be a universal and promising tool whose potential is not yet fully understood. The material obtained in this process is very simple to modify, and, at the same time, it is a strong structure that achieves results similar to native tissue. The decellularized tissue is not only an immunologically neutral scaffold, but also, due to the high content of extracellular molecules, has an active effect on the metabolism of the surrounding cells. Decellularization promotes regeneration of the injured tissue. The native molecular structure of the scaffold promotes integration of healing tissues and also preserves mechanical resistance, preventing reinjuries, which are a common complication at the tendon, ligament or fascia healing process. It shows superior regenerative potential compared to commercially available synthetic scaffolds but also faces the challenges. 

Multiple methods of decellularization have been developed. However, protocols combining multiple methods show greater potential than single-component procedures. Using several components, with other mechanisms of action, enlarges the decellularization effect, but also allows minimization of the doses of various agents. The destructive effect of decellularization is intensified when applying higher amounts of single components. In addition, crucial to operating with the physical decellularization methods seems to be one of its abilities, that of loosening the collagen fibers without dissolving them, which not only improves cell extraction, but also promotes and expedites cell reseeding. The development of a simple and efficient decellularization protocol will make it possible to automate this process as quickly as possible and make it widely available in various forms, such as in universal DRCT implants, wound dressing, hydrogels or filaments for 3D bioprinting.

Many experiments remain necessary to fully understand the potential of decellularization and, above all, to analyze and optimize currently available solutions. However, the current techniques provide promising results for therapeutic use.

## Figures and Tables

**Figure 2 cells-12-02293-f002:**
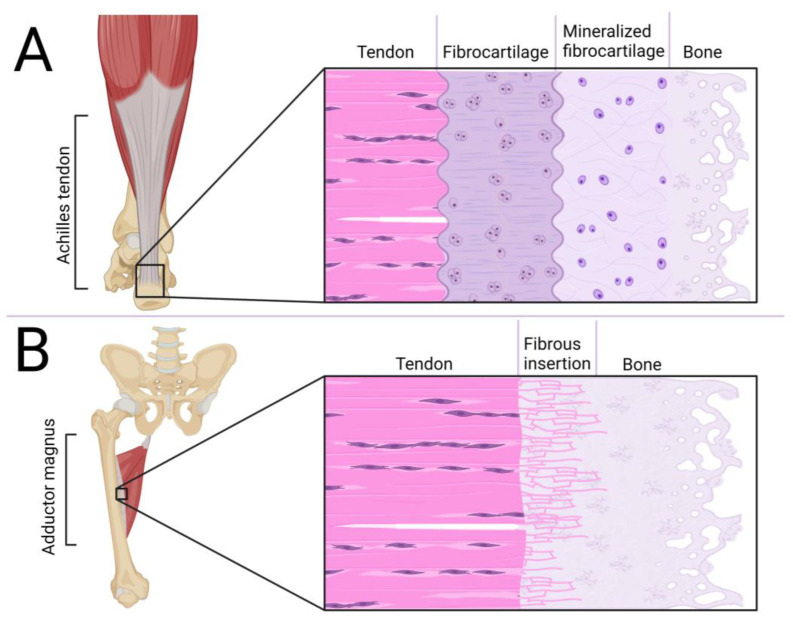
Comparison of fibrocartilaginous (**A**) and fibrous (**B**) enthesis.

**Figure 3 cells-12-02293-f003:**
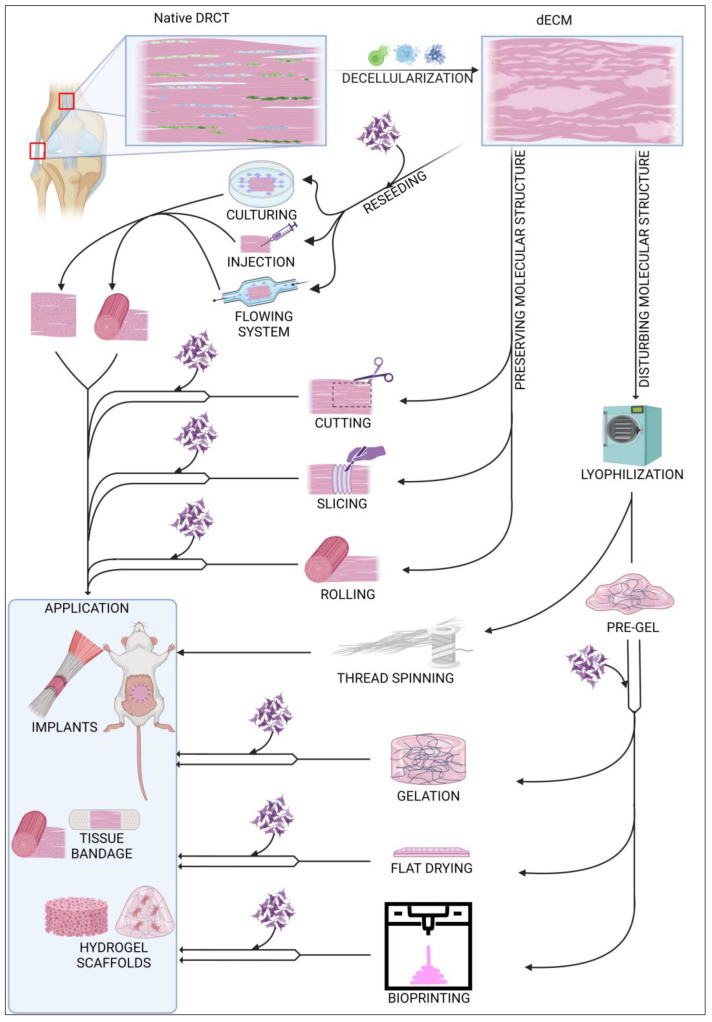
Summary of commonly used modifications of decellularized ECM.

**Table 1 cells-12-02293-t001:** Expression of fibroblasts markers in different types of cells. VIM, vimentin; PDGFRA, platelet-derived growth factor receptor alpha; FSP-1, fibroblast-specific protein I; COL1, collagen type I alpha 1; MMP-1, matrix metalloproteinase-1; SCX, scleraxis; +, presence of fibroblast marker; −, absence of fibroblast marker; NA, no data available.

	Tenocytes	Ligament Fibroblasts	Fasciacytes	Fascial Fibroblasts
VIM	+	+	+	+
	[36]	[37]	[27]	[27]
PDGFRA	+	+	NA	+
	[1]	[38]		[39]
FSP-1	+	+	+	−
	[40]	[41]	[27]	[27]
COL1	+	+	NA	+
	[42]	[41]		[43]
MMP-1	+	+	NA	+
	[44]	[41]		[45]
SCX	+	+	NA	NA
	[40]	[46]		

**Table 2 cells-12-02293-t002:** Commonly used methods of decellularization in DRCT tissue.

	Method	Mechanism of Action	Efficiency	Effect on ECM	Comments	Source
Physical	Electroporation	Poration of cell membrane through destabilizing its electric potential	Insufficient	Forms microcavities that do not affect the strength of sample	Can be processed in situ	[59,60,61]
	Sonication	Disrupting cell membrane by high-frequency sound waves	Low	Loosening of the collagen fibers, but used with lower intensity preserves ECM structure	Lack of information about using it in DRCT	[71,72]
	Freeze–Thaw	Disruption of cell membranes by freezing ice crystals	High	Preserves native structure	Rinsing sample even with distilled water increase the effectiveness of procedure	[63,73,74]
	High Hydrostatic Pressure	Loosening of the phospholipid bilayer	High	Preserves native structure	There is a lack of information about efficiency in DRCT	[75,76]
Chemical	Triton X-100	Disruption of lipid bilayer of cell membrane	Insufficient	Loosening of the fibers, but using low concentration can preserve the ECM structure	The most commonly used procedure in DRCT	[77,78,79]
	SDS	Disruption of covalent bonds between membrane proteins	High	Dissolves the fibers leading to merging into a homogeneous mass	Has a long-lasting cytotoxic effect and requires advanced washing	[80,81]
	Sodium Azide	Inhibition of cytochrome C oxidase	High	Preserves native structure	Commonly used bacteriostatic agent	[82,83]
	Latrunculin	Depolymerization of actin filaments to destroy cell cytoskeleton	High	Preserves native structure	There is a lack of information about efficiency in DRCT	[84,85]
	EDTA	Reducing cell adhesion to ECM by binding to metal cations	Low	Reduces the number of proteins in some samples	Commonly used as supplement increasing trypsin activity	[86,87]
	Trypsin	Digesting membrane proteins leading to membrane permeability	High	Dissolves the fibers, but used with low concentration preserves the native structure	Demonstrates high efficiency even in low concentration	[88,89]
	Nucleases	Cleaving DNA or RNA released from the cells disrupted by other agents	Low	Preserves native structure	As supplement to protocol, greatly increases efficiency of other methods	[90,91]

## Data Availability

Not applicable.

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
