# Peer review of "Decellularization of Dense Regular Connective Tissue—Cellular and Molecular Modification with Applications in Regenerative Medicine"

_cells, 2023, doi:10.3390/cells12182293_

Round 1

Reviewer 1 Report

Dear author,

Thank you for giving me the opportunity to review your work.

It deals with reviewing the data and knowledge regarding the decellularization of dense regular connective tissue. The goal is to demonstrate that cellular and molecular modifications are  compatible with applications in regenerative medicine. This work is very useful regarding the type of tissue concerned, for which data are not currently well organized.

I have few comments and questions on this work:

- It could be interesting to explain the process of DRCT healing after injury. What are the different steps, which molecules and cells involved? is there a scar? Can you link your answer to the regeneration process?

- Is deoxycholic acid also used for decellularisation of DRCT and if not, why?

- Same question for sodium azide.

- To assess the efficiency and safety of decellularization, you should have previously described the criteria of success with quantitative information. Can you provide this for DRCT? Also explain differences in these criteria between autologous, heterologous and xenogenic grafting.

- One of the main issues in this kind of review is to deal with the big variability within methods used to characterize the obtained product, i.e. biological and mechanical properties obviously. Can you address this issue and to which extent the results you comment are comparable?

- For chemical methods, concentration, duration and possibly  stirring may have an impact on their success. Any complementary information on that?

- Can you tell what kind of cells are commonly used/or the most promising for recellularization of the dECM? 

- Can cell differentiation be a goal in this strategy?

- How long is the remodeling process suitable?

Thank ou in advance for your answers

Sincerely

The text may be improved, but there is no big issue.

Reviewer 2 Report

It is a well-written concise review describing dense regular connective tissue.

First, the authors discussed the essential morphological and molecular features of DRCT in detail, such as SOX9 and GLI1, as regulatory genes or specific cell markers. This is interesting; however, they did not clearly discuss them in the later section, especially how and why the decellularized tissue could maintain or enhance these essential features, as the title suggests.

I would also recommend adding "high hydrostatic pressure" as one of the mechanical methods for decellularization.

Taken together, the manuscript requires some modification before publication.

Reviewer 3 Report

This review is a summary of the current knowledge of DRCT decellularization. The manuscript presents a balanced update of what is currently known on the topic, that tissue engineering researchers and orthopedic surgeons are waiting for a comprehensive compilation. This review article is very useful for new researchers to keep abreast of developments in the field. The advantages as well as disadvantages of each technique are covered. Complementary techniques are also carefully described in Section 4.5. Overall, the explanations are, for the most part, convincing. This manuscript is appropriately cited in the literature.

Round 2

Reviewer 1 Report

Dear author,

Thank you for your clear answers to my questions and comments.

I agree with them.

Best regards